# Extended Postoperative Analgesia via Caudal Catheters for Major Surgery in Neonates—A 6-Year Retrospective Study

**DOI:** 10.3390/jcm14082651

**Published:** 2025-04-12

**Authors:** Stefan Heschl, Brigitte Messerer, Corinna Binder-Heschl, Michael Schörghuber, Maria Vittinghoff

**Affiliations:** 1Division of Anesthesiology and Intensive Care Medicine 1, Department of Anesthesiology and Intensive Care Medicine, Medical University of Graz, 8047 Graz, Austria; brigitte.messerer@medunigraz.at (B.M.); maria.vittinghoff@medunigraz.at (M.V.); 2Division of Neonatology, Department of Pediatrics, Medical University of Graz, 8047 Graz, Austria; corinna.binder@medunigraz.at; 3Division of Anesthesiology and Intensive Care Medicine 2, Department of Anesthesiology and Intensive Care Medicine, Medical University of Graz, 8047 Graz, Austria; michael.schoerghuber@medunigraz.at

**Keywords:** neonate, caudal anesthesia, catheter, epidural morphine

## Abstract

**Background**: Caudal anesthesia is an important regional anesthetic technique in neonates. The placement of a catheter can provide excellent analgesia for a prolonged period; the role of adjuvants, in particular morphine, however, remains unclear. We aimed to describe our experience with caudal catheters for major surgery in neonates. **Methods**: We included all neonates who had a caudal catheter placed for major abdominal and thoracic surgery and explored postoperative pain management and catheter complications. This retrospective case series included neonates with caudal catheter placement from October 2012 to April 2018 at a tertiary university hospital. **Results**: A total of 33 caudal catheter placements in 32 neonates were included in this study, of which 28 (85%) were a laparotomy and 5 (15%) a thoracotomy. The mean catheter duration was 135 h with a postoperative failure rate of 3%. Patients who did not receive intravenous opioids postoperatively had a significantly shorter stay in the intensive care unit than those who did (341 h vs. 674 h, *p* = 0.01). All patients received continuous local anesthetics over the catheter, and 79% received additional intermittent epidural morphine postoperatively for a median period of 42 h. No infectious complications were reported. **Conclusions**: Caudal catheters are a valuable option for perioperative analgesia for major surgery in neonates. We found no serious catheter-related complication. Further research is needed to define the optimal approach and combination of different analgesic techniques.

## 1. Introduction

Caudal anesthesia, the injection of local anesthetic into the epidural space via the sacral hiatus, is one of the most important regional anesthetic techniques in pediatric anesthesia. Its widespread use among pediatric anesthetists can be attributed to the safety and simplicity of the procedure [1,2]. It can be used in combination with general anesthesia or as the sole anesthetic technique for any surgical procedure below the umbilicus. By adding morphine to the caudal injection, however, even upper abdominal and thoracic procedures can be covered due to the cephalad spread of the hydrophilic opioid [3]. A single injection of morphine induces analgesia for more than 12 h [4]. For major abdominal or thoracic surgery in neonates, where significant pain for longer than 12 h can be expected, the placement of a catheter in the epidural space via the caudal approach offers the benefit of continuous or repeated drug application for analgesia in the postoperative period. Despite this advantage, only 1% of patients receiving a caudal block had a catheter placed in a large multicenter survey [5]. This might in part be due to the fact that positioning of the catheter at the desired level is technically difficult [6,7]. The position of the catheter tip is probably not as important when morphine is administered compared to local anesthetic alone due to the cephalad distribution of hydrophilic morphine. However, not much has been published about the ideal combination of caudally administered morphine and local anesthetic, and there is a wide variety of practices [2]. Furthermore, there is a lack of evidence about the repeated bolus dosing of epidural morphine. We therefore aimed to investigate our practice with caudal catheters for major abdominal and thoracic surgery in neonates and describe postoperative pain management, including intermittent epidural morphine and possible effects on the postoperative course and complications of long-term catheter use.

## 2. Materials and Methods

This retrospective data analysis was carried out at the University Hospital Graz, a tertiary center with a dedicated pediatric anesthesia section.

In this case series, we included all neonates from birth up to 28 days of life from both sexes who had a caudal catheter placed for a major surgical procedure (laparotomy or thoracotomy) and had a postoperative admission to the surgical pediatric intensive care unit (PICU). At our institution, caudal catheters are placed in all neonates undergoing major thoracic or abdominal surgery, including emergency surgery, unless there is a clear contraindication. The study period was October 2012 (start of our electronic documentation system) until April 2018. All caudal catheters were placed by specialized pediatric anesthetists using landmark guidance without routine tip position confirmation under strict aseptic conditions in the operating theater. The desired insertion length was approximated on the patient’s back, and all catheters were tunneled subcutaneously away from the perianal region. A total of 2–3.75 mg/kg (1 mL of either 0.2% or 0.375%) of ropivacaine and 30–50 µg/kg of morphine were given as initial bolus via the caudally threaded catheter to all patients. A postoperative pain assessment was performed regularly every 4 h and additionally when needed using the Infants Postoperative Pain Scale with an intervention threshold of 4 points [8]. Pain management followed a standardized institutional stepwise multimodal protocol. This included the continuous infusion of metamizol and on demand intravenous morphine for all patients. In addition to a continuous infusion of ropivacaine 0.2% via an automated infusion pump following the published recommendations [9], 30–50 µg/kg of morphine was administered manually if the intervention threshold was reached, at most every 12 h. The continuous infusion of ropivacaine was eventually paused at the discretion of the PICU physician. Pain and the need for additional analgesics were continuously assessed, and the catheter was removed when considered clinically appropriate.

Eligible patients were identified from the electronic patient record of the anesthesia documentation system. This list was crosschecked against the electronic patient records from the PICU. Relevant data were extracted from both databases (anesthesia and PICU electronic patient record) for the statistical analysis. The postoperative parameters that were explored in this study included epidural morphine use, intravenous opioid use, duration of ventilation, time to first stool, time to enteral nutrition, and length of PICU stay (PICU-LOS). The microbiology test reports and individual patient histories were manually screened in the electronic hospital patient record to detect any complications possibly associated with the caudal catheter.

Data analysis was conducted using SPSS version 25^®^ (IBM, Chicago, IL, USA). Categorical data are presented as frequencies and proportions. All continuous variables were tested for normality using the Shapiro–Wilk test. Depending on their distribution, they are presented as median (interquartile range [IQR]) or mean (standard deviation [SD]). Due to the small sample size, no inferential statistics were used.

## 3. Results

### 3.1. Patient Characteristics

A total of 32 neonates were included in this study, of which one neonate had a catheter placed for bowel resection because of necrotizing enterocolitis on day three of life and after planned removal had another caudal catheter placed for a surgical revision of the abdomen on day eleven. This gives a sum of 33 caudal catheter placements, of which 28 (85%) were a laparotomy and 5 (15%) a thoracotomy. Patient characteristics are described in Table 1.

Twenty-eight caudal catheters were placed in the first attempt and three in the second (missing data for two patients). One catheter was removed after 100 h because it was blocked; all other catheters were removed at the time they were considered not necessary anymore by the treating pediatric intensive care physician. This results in a postoperative failure rate of 3%. Other than this blockage, neither major nor minor catheter-related complications were reported. Postoperative ventilation was required in 21 patients (64%) for a mean of 77 h (SD 50 h). In four patients, the catheter was removed while the patients were still ventilated, and in all other cases, the catheter was removed some time after extubation. The caudal catheters were left in place for a mean of 135 h (SD 63 h), and the longest catheter duration was 260 h (Figure 1).

### 3.2. Postoperative Epidural Morphine

At least one dose of epidural morphine (30–50 µg/kg) was administered to 26 patients (79%) postoperatively on the PICU (Table 2). Intermittent re-dosing (at most 12 hourly) was performed for a median of 42 h (IQR 26–129).

### 3.3. Additional Intravenous Opioid

Of note, six patients did not receive any intravenous opioid after completion of the surgery. Those patients had a shorter median PICU (341 h vs. 674 h) (Table 3). Two patients who did not receive intravenous opioids postoperatively did not receive epidural morphine either.

### 3.4. Type of Surgery

Patients who had a caudal catheter placed for a thoracotomy had a longer duration of surgery (194 vs. 101 min) (Table 4).

### 3.5. Microbiology

None of the 33 catheters were removed because of a clinical suspicion of infection. Due to institutional policy, however, all catheter tips were sent for routine microbiological testing after removal. Twenty-eight specimens showed no signs of any bacterial growth. There were three cases of Staphyloccus epidermidis, one Bacillus species, and one Micrococcus luteus growth. However, all five cases (15%) were considered either as colonization or contamination, and none were considered to be a clinically relevant infection by the treating physicians. No specific antimicrobial therapy had to be initiated.

## 4. Discussion

In a retrospective review of 33 caudally placed epidural catheters in neonates, we found a mean catheter duration of 135 h with a postoperative failure rate of 3%. No clinically relevant catheter-related infection was detected, even after prolonged catheter use. Six patients (18%) did not receive any intravenous opioid after surgery.

While single shot caudal block remains a mainstay of pediatric regional anesthesia, the placement of a catheter for prolonged postoperative analgesia is rarely performed [5]. The most likely explanation for this low rate of catheter placement is the lack of a need for prolonged analgesia in typically minor procedures such as circumcision or inguinal hernia repair where a caudal block is used as the anesthetic technique. For major surgery, with expected extensive postoperative pain, however, placement of a catheter offers the possibility to prolong the duration of analgesia well beyond that of a single shot block, even when additives are used. The placement of a caudal catheter and, in particular, ensuring accurate tip location for higher lumbar or thoracic levels are time-consuming and technically difficult. Various techniques, including X-ray-based techniques such as epidurograms and fluoroscopy, electrical stimulation, and ultrasound, have been described; however, these techniques are only used in half of all patients, possibly due to the added complexity [10]. It needs to be highlighted, however, that catheter tip location using ultrasound might change practice due to its simplicity and an increasing familiarity of pediatric anesthetists with ultrasound, but there is further need for investigation of the clinical benefits of this technique [11,12,13]. In our setting, confirmation of tip location was not routinely performed, making the process of catheter placement easier and possibly less time-consuming. Epidural morphine, due to its cephalad spread as a hydrophilic opioid, is able to reach higher sensory levels, and even the brain stem, and produces long-term (median 12 h) analgesia in children [14,15]. Therefore, when adding epidural morphine, the exact catheter tip location is not as important compared to using local anesthetic alone. Importantly, possible postoperative catheter tip position changes do not affect analgesic efficacy as much. Another possible advantage of epidural morphine is the reduced or absent need for intravenous opioids such as systemic analgesia. This could potentially enhance weaning and reduce ventilation time. Prospective, randomized trials for postoperative, epidural morphine are lacking; however, retrospective data suggest sufficient pain control in infants with limited data for neonates [16]. Assessing pain in a ventilated neonate and differentiating it from any other distress, however, are clinically challenging. Systemic opioids are therefore often prescribed rather liberally. In our series, six patients with laparotomies were managed without systemic opioids postoperatively, and two of these patients did not receive epidural morphine postoperatively either. It remains unclear from our small dataset whether the almost halved PICU-LOS was at least in part due to the avoidance of systemic opioids or whether a positive clinical course rendered systemic opioid analgesia unnecessary. Our findings do suggest, however, that intravenous opioids are not automatically necessary in the postoperative period after major abdominal and thoracic surgery in neonates when caudal anesthesia with added epidural morphine was used as the anesthetic technique. In a randomized trial of 30 neonates, caudal catheters, even without the addition of epidural morphine, led to lower postoperative pain scores and significantly reduced the duration of mechanical ventilation [17], which is in line with the results from our retrospective study. It needs to be added, however, that intravenous morphine increases the tolerance of mechanical ventilation, which might be a separate indication for its use. Also, the respiratory depressant effect of epidural compared to intravenous morphine cannot be calculated from our data.

Another reason to avoid catheter placement is the fear of infection. While a large registry study found no infectious complication associated with single shot blocks, risk of infection constituted a typical complication of catheter use, and it increased by about 6.7% for every additional day the catheter remained in place [18]. Apart from only one epidural abscess, however, the authors found only minor local cutaneous infections in their study. In our investigation, no patient showed signs of a catheter-related infection. This finding indicates that even for prolonged catheter use in neonates, the benefits of regional anesthesia possibly outweigh the microbiological risks. This is also in accordance with previous findings for short-term use (3 days) in another large registry dataset; however, our colonization rate of 15% is even lower than that reported in this study [19]. All of our catheters were tunneled, which has already been shown to reduce bacterial colonization to a comparable rate (11%), as in our study, and should be encouraged [20]. Strict aseptic handling during placement, as well as in the postoperative period, seems prudent in reducing infectious complications. All but one of the catheters in our study were removed following the standardized catheter weaning protocol, resulting in mean removal on postoperative day six but with a wide range from postoperative day one to ten. This suggests that an individualized approach regarding time of catheter removal in this selected patient group seems clinically beneficial; however, we are unable to conclude that with certainty from our data. Even the catheter that had to be removed after 100 h due to blockage had provided substantial analgesia up to that point and could therefore still be considered clinically helpful for about 4 days. Furthermore, it can be concluded that the catheters in our patients were providing clinically meaningful analgesia for a prolonged period; however, this relied somewhat on the clinical interpretation of the treating PICU physician. We suggest evaluating different postoperative pain regiments in future prospective trials.

### Limitations

A major limitation of this study is its retrospective nature without a control group. Even though the study period was 6 years at a major tertiary center with a department for pediatric surgery, the sample size is still small. Therefore, our results need to be interpreted cautiously. The results of this study, however, will add to the body of evidence regarding regional anesthesia in neonates for major thoracic and abdominal surgery as prospective studies on ideal (regional) anesthetic management in such patients are unlikely for many reasons. The number of neonates undergoing major thoracic or abdominal surgery not receiving a caudal catheter due to contraindications during the study period could not be determined. However, since all neonates, including systemically unwell patients, for emergency surgery receive a caudal catheter when strict contraindications are absent, we suspect this number to be very low. Even though we followed a structured stepwise institutional protocol for postoperative pain, clinical interpretation plays an important role in neonates. As it is clinically difficult sometimes to discern pain from distress due to other reasons, postoperative pain management might be influenced by other factors not investigated in this study. We did not record the time it took to place the catheter; however, our clinical experience and also the low number of necessary attempts suggest that placing a caudal catheter where exact positioning of the tip is not performed is rather straightforward. There is no standardized documentation of specific complications in our electronic patient record, so even though all patient histories were manually screened, it is possible that minor complications which were not documented were missed.

## 5. Conclusions

In conclusion, we found that the administration of epidural morphine via a caudally placed epidural catheter provided clinically adequate analgesia for the postoperative period after major abdominal and thoracic surgery in neonates. We found no major complication associated with caudal catheters in neonates. We conclude that caudally inserted epidural catheters are a valuable part of perioperative analgesia for major thoraco-abdominal surgery in neonates.

## Figures and Tables

**Figure 1 jcm-14-02651-f001:**
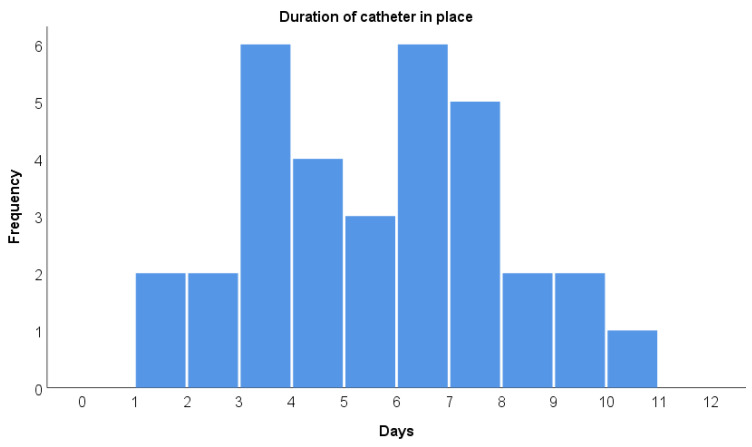
Duration of catheter in place.

**Table 1 jcm-14-02651-t001:** Patient characteristics.

Age [days]	2 (1–8) {0; 27}
Gestational age at time of operation [days]	260 (247–277) {222; 297}
Weight [g]	2700 (2100–2900) {1400; 4200}
Male [n]	17 (52%)
Thoracotomy [n]	5 (15%)
Laparotomy [n]	28 (85%)
Duration of surgery [minutes]	107 (67–162)
Epidural morphine postoperatively [n]	26 (79%)

Data are presented as median (IQR) {minimum, maximum} or n (proportion) as appropriate.

**Table 2 jcm-14-02651-t002:** Postoperative epidural morphine vs. no epidural morphine.

	Postoperative Epidural Morphine n = 26	No Postoperative Epidural Morphine n = 7
Age [days] (median, IQR)	3 (1–8)	2 (0–6)
Gestational age [days] (mean, SD)	261 (19)	259 (17)
Weight [g] (mean, SD)	2600 (700)	2800 (800)
Male [n]	14 (54%)	3 (43%)
Duration of surgery [minutes] (mean, SD)	102 (56)	166 (62)
Duration of catheter [hours] (mean, SD)	140 (59)	123 (52)
Type of surgery		
Laparotomy [n]	23 (89%)	5 (71%)
Thoracotomy [n]	3 (11%)	2 (29%)
i.v. opioid [n]	22 (85%)	5 (71%)
Duration of i.v. opioid [hours] (median, IQR)	29 (16–70)	25 (0–120)
Total i.v. morphine [mg/kg] (median, IQR)	0.2	1.0
Postop ventilation [n]	16 (62%)	5 (71%)
Duration of ventilation [hours] (mean, SD)	71 (52)	71 (32)
Time to first stool [hours] (mean, SD)	41 (33)	35 (19)
Time to enteral nutrition [hours] (median, IQR)	30 (21–73)	29 (25–46)
PICU LOS [hours] (median, IQR)	671 (478–739)	420 (287–1640)

**Table 3 jcm-14-02651-t003:** Postoperative intravenous opioid vs. no opioid.

	No i.v. Opioid n = 6	i.v. Opioid n = 27
Age [days] (median, IQR)	4 (1–12)	2 (1–8)
Gestational age [days] (mean, SD)	273 (18)	257 (17)
Weight [g] (mean, SD)	3200 (500)	2500 (700)
Male [n]	2 (33%)	15 (56%)
Type of surgery		
Laparotomy [n]	6 (100%)	22 (82%)
Thoracotomy [n]	0 (0%)	5 (18%)
Duration of surgery [minutes] (mean, SD)	111 (62)	116 (63)
Duration of catheter [hours] (mean, SD)	98 (49)	145 (56)
Epidural morphine [n]	4 (67%)	22 (82%)
Duration of epidural morphine [hours] (median, IQR)	3 (0–43)	38 (18–105)
Postop ventilation [n]	2 (33%)	19 (70%)
Duration of ventilation [hours] (mean, SD)	39 (31)	74 (48)
Time to first stool [hours] (mean, SD)	23 (25)	44 (30)
Time to enteral nutrition [hours] (median, IQR)	24 (13–66)	30 (24–72)
PICU LOS [hours] (median, IQR)	341 (182–530)	674 (505–1058)

**Table 4 jcm-14-02651-t004:** Type of surgery.

	Laparotomy n = 28	Thoracotomy n = 5
Age [days] (median, IQR)	4 (1–8)	1 (1–2)
Gestational age [days] (mean, SD)	261 (19)	260 (16)
Weight [g] (mean, SD)	2500 (600)	3300 (1000)
Male [n]	13 (46%)	4 (80%)
Duration of surgery [minutes] (mean, SD)	101 (55)	194 (43)
Duration of catheter [hours] (mean, SD)	141 (57)	110 (59)
Epidural morphine [n]	23 (82%)	3 (60%)
Duration of epidural morphine [hours] (median, IQR)	35 (7–100)	27 (0–87)
i.v. opioid [n]	22 (82%)	5 (100%)
Duration of i.v. opioid (median, IQR)	26 (9–72)	66 (30–95)
Postop ventilation [n]	17 (61%)	4 (80%)
Duration of ventilation [hours] (mean, SD)	72 (48)	63 (50)
Time to first stool [hours] (mean, SD)	39 (32)	41 (1)
Time to enteral nutrition [hours] (median, IQR)	30 (23–74)	26 (21–58)
PICU LOS [hours] (median, IQR)	649 (465–742)	406 (296–1298)

## Data Availability

The data presented in this study are available on request from the corresponding author.

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
