# Peer review of "Extended Postoperative Analgesia via Caudal Catheters for Major Surgery in Neonates—A 6-Year Retrospective Study"

_jcm, 2025, doi:10.3390/jcm14082651_

Round 1
Reviewer 1 Report
Comments and Suggestions for Authors
Thank you for submitting you research to the Journal of Clinical Medicine
Retrospective cohort studies such as this are useful references for clinicians when benchmarking institutional practice and considering change.
Overall the paper is well written and clearly presented.
My general questions and comments:
- As a retrospective review of a small number of interventions over a significant time period, I suppose (as it is not clearly stated) that there is no single primary outcome. I think it would be useful to state this in both the abstract and the methodology sections. It is harder to read study without a primary outcome when this is not clearly stated.
- I would recommend deleting the inferential p value statistics from such small groups in a retrospective study, and reference to the statistical significance between the groups. Furthermore, the only four "significant" p value findings are not clinically informative or conclusive, and discussing these points (duration of surgery and epidural morphine, then patient weight and PICU LOS versus IV opioids, duration of surgery for thoracotomy versus laparotomy) does not add to the paper's quality of demonstrating one institution's confidence and success with caudal catheters.
Specific comments:
- The abstract is a little long. One obvious place to trim is abstract lines 22/23: Replace "No infectious complication was found, however in 5 patients (15%) bacterial colonization or contamination of the catheter was encountered" with "No infectious complications were reported".
- Neonates having laparotomies are often systemically unwell, with presumed or confirmed sepsis. It is unclear whether caudal catheters are considered in this patient group in your institution, or whether they were employed in relatively well, sepsis free patients (malrotations, duplications etc). Please clarify.
- In the same vein, it is completely unclear what proportion of all neonates received caudal catheters for laparotomies and how this was decided. What is your reference population size? How many neonatal laparotomies and thoracotomies are performed (even roughly), and how is it decided whether a caudal catheter is used, or not
- Materials and methods, lines 70/71: Ropivacaine 0.2% infusion rates are not elucidated, and this seems an important piece of information. Very low infusion rates in smaller sized neonates may lead to logistical issues from and concerns regarding toxicity. Indeed, NYSORA recommends limiting ropivacaine infusions to 48 hours for these reasons, yet your institution continues infusions for a mean of over five days. Please explain. https://www.nysora.com/topics/sub-specialties/pediatric-anesthesia/pediatric-epidural-spinal-anesthesia-analgesia/#:~:text=Cumulative%20toxicity%20is%20a%20concern,the%20older%20local%20anesthetic%20bupivacaine.
- Limitations: If you haven't been able to determine what proportions of neonates receive caudal catheters and why, this is very significant. Is it anaesthesia operator dependent?
- Limitations 236-8: Prospective randomised studies of of regional anaesthesia in such patient groups are unlikely for many reasons, as I am the authors are aware. It may be more truthful to state that further observational studies will add to the body of evidence regarding regional anaesthesia in such patients.
Author Response
Please see the attachment.
Response to Reviewer 1
Comments and Suggestions for Authors
Thank you for submitting you research to the Journal of Clinical Medicine
Retrospective cohort studies such as this are useful references for clinicians when benchmarking institutional practice and considering change.
Overall the paper is well written and clearly presented.
Thank you for reviewing our manuscript and your constructive comments.
My general questions and comments:
- As a retrospective review of a small number of interventions over a significant time period, I suppose (as it is not clearly stated) that there is no single primary outcome. I think it would be useful to state this in both the abstract and the methodology sections. It is harder to read study without a primary outcome when this is not clearly stated.
You are right that there was no single primary outcome and we have added this useful information for the reader in both the abstract and the methodoloy section (Lines 13, 86-87)
2. I would recommend deleting the inferential p value statistics from such small groups in a retrospective study, and reference to the statistical significance between the groups. Furthermore, the only four "significant" p value findings are not clinically informative or conclusive, and discussing these points (duration of surgery and epidural morphine, then patient weight and PICU LOS versus IV opioids, duration of surgery for thoracotomy versus laparotomy) does not add to the paper's quality of demonstrating one institution's confidence and success with caudal catheters.
We have followed your suggestions and removed the p-values and references to significant differences from the manuscript. (Table 2,3,4, Lines 140, 146, 171) A sentence why no inferential statistics were performed was added to the methods section. (Lines 95-96). Furthermore, we have removed the discussion points about clinically not informative findings as suggested. (Lines235, 237)
Specific comments:
- The abstract is a little long. One obvious place to trim is abstract lines 22/23: Replace "No infectious complication was found, however in 5 patients (15%) bacterial colonization or contamination of the catheter was encountered" with "No infectious complications were reported".
The abstract was shortened according to your recommendation. (Line 13, 22)
2. Neonates having laparotomies are often systemically unwell, with presumed or confirmed sepsis. It is unclear whether caudal catheters are considered in this patient group in your institution, or whether they were employed in relatively well, sepsis free patients (malrotations, duplications etc). Please clarify.
Systemically unwell patients undergoing emergency surgery were also included and routinely receive caudal catheters, unless there is a clear contraindication. (Lines 63-65, 278-283)
3. In the same vein, it is completely unclear what proportion of all neonates received caudal catheters for laparotomies and how this was decided. What is your reference population size? How many neonatal laparotomies and thoracotomies are performed (even roughly), and how is it decided whether a caudal catheter is used, or not
At our institution, caudal catheters are placed in all neonates undergoing major thoracic or abdominal surgery, unless there is a strict contraindication. Therefore only very few patients did not receive a caudal catheter. This is now explained in the manuscript. (Lines 63-65, 278-283)
4. Materials and methods, lines 70/71: Ropivacaine 0.2% infusion rates are not elucidated, and this seems an important piece of information. Very low infusion rates in smaller sized neonates may lead to logistical issues from and concerns regarding toxicity. Indeed, NYSORA recommends limiting ropivacaine infusions to 48 hours for these reasons, yet your institution continues infusions for a mean of over five days. Please explain. https://www.nysora.com/topics/sub-specialties/pediatric-anesthesia/pediatric-epidural-spinal-anesthesia-analgesia/#:~:text=Cumulative%20toxicity%20is%20a%20concern,the%20older%20local%20anesthetic%20bupivacaine.
We follow published ASRA/ESRA guidelines on Ropivacaine dosing for continous epidural infusion(Suresh S, Ecoffey C, Bosenberg A, et al. The European Society of Regional Anaesthesia and Pain Therapy/American Society of Regional Anesthesia and Pain Medicine Recommendations on Local Anesthetics and Adjuvants Dosage in Pediatric Regional Anesthesia. Reg Anesth Pain Med. 2018;43(2):211-216. doi:10.1097/AAP.0000000000000702. This explanation and reference was added to the manuscript. (Line 77)
5. Limitations: If you haven't been able to determine what proportions of neonates receive caudal catheters and why, this is very significant. Is it anaesthesia operator dependent?
At our institution, caudal catheters are placed in all neonates undergoing major thoracic or abdominal surgery, unless there is a contraindication. Therefore only very few patients did not receive a caudal catheter. This is now explained in the manuscript. (Lines 63-65, 278-283)
6. Limitations 236-8: Prospective randomised studies of of regional anaesthesia in such patient groups are unlikely for many reasons, as I am the authors are aware. It may be more truthful to state that further observational studies will add to the body of evidence regarding regional anaesthesia in such patients.
This has been changed in the manuscript. (Lines 276-278
Reviewer 2 Report
Comments and Suggestions for Authors
I appreciate the authors' presentation of their experience with caudal epidural catheters for postoperative analgesia in neonates at their institution. From my point of view, the fear of infection could be the main reason for pediatric anesthetists being reluctant to place caudal epidurals. As the authors pointed out, the small number of patients limits the ability to draw firm conclusions from this retrospective study.
The authors examined 32 neonates who underwent major abdominal and thoracic surgery and received caudal catheters for postoperative pain management. It would be beneficial for clinical practice if the authors could compare postoperative outcomes between neonates with and without caudal catheters. Furthermore, a comparison of postoperative outcomes between neonates with caudal catheters and those receiving a single caudal block would be even more valuable.
Author Response
Please see the attachment.
Response to Reviewer 2
Comments and Suggestions for Authors
I appreciate the authors' presentation of their experience with caudal epidural catheters for postoperative analgesia in neonates at their institution.
Thank you for reviewing our manuscript and your constructive comments.
From my point of view, the fear of infection could be the main reason for pediatric anesthetists being reluctant to place caudal epidurals.
We agree and have therefore emphasized the microbiological safety endpoints we have found in our study.
As the authors pointed out, the small number of patients limits the ability to draw firm conclusions from this retrospective study.
We agree and have therefore removed the inferential statistics as also suggested by another reviewer, due to the small sample size.
The authors examined 32 neonates who underwent major abdominal and thoracic surgery and received caudal catheters for postoperative pain management. It would be beneficial for clinical practice if the authors could compare postoperative outcomes between neonates with and without caudal catheters. Furthermore, a comparison of postoperative outcomes between neonates with caudal catheters and those receiving a single caudal block would be even more valuable.
At our institution, caudal catheters are placed in all neonates undergoing major thoracic or abdominal surgery, unless there is a strict contraindication. Therefore only very few patients did not receive a caudal catheter. Unfortunately it is therefore impossible to make such a comparison from our dataset. This is now explained in the manuscript. (Lines 63-65, 278-283)
Round 2
Reviewer 2 Report
Comments and Suggestions for Authors
The replies presented by the authors are acceptable.
Author Response
The replies presented by the authors are acceptable.
Thank you again for reviewing our manuscript.